# The Impact of Home Parenteral Nutrition on Survival and Quality of Life in Patients with Intestinal Failure and Advanced Cancer: A Comprehensive Review

**DOI:** 10.3390/nu17050905

**Published:** 2025-03-05

**Authors:** Miroslaw Perlinski, Jacek Sobocki

**Affiliations:** 1Fresenius Kabi Poland, Aleje Jerozolimskie 134, 02-305 Warsaw, Poland; 2Department of General Surgery, and Clinical Nutrition, Medical Center of Postgraduate Education, Czerniakowska 231, 00-416 Warsaw, Poland; sobockij@gmail.com

**Keywords:** home parenteral nutrition, chronic intestinal failure, quality of life, survival, oncology, malignant bowel obstruction, nutrition

## Abstract

**Background and Aims:** Home parenteral nutrition (HPN) is essential in the management of chronic intestinal failure (CIF) and malignant bowel obstruction (MBO), particularly in cases where enteral feeding is not feasible. This review examines the evidence from 34 studies to evaluate the impact of HPN on survival and quality of life (QoL) in patients with MBO, CIF, and advanced cancer, as well as to identify clinical predictors of survival and address psychosocial challenges. **Methods:** A comprehensive review was conducted of 34 studies, focusing on the use of HPN in patients with MBO, CIF, and advanced cancer. Data were analyzed for survival outcomes, QoL metrics, and predictors of survival, including performance status, albumin levels, and the continuation of chemotherapy. The psychosocial aspects of HPN therapy were also assessed, particularly its influence on patient’s daily lives and mental health. **Results:** Key predictors of improved survival included good performance status, higher albumin levels, and the ability to continue chemotherapy. While HPN extended survival in many cases, its impact on QoL varied significantly. Psychosocial challenges, including lifestyle disruption and mental health strain, were common among long-term HPN patients, underscoring the need for comprehensive patient support. **Conclusions:** HPN can be a life-sustaining therapy for patients with CIF, MBO, and advanced cancer, but its success depends on careful patient selection and management. Identifying predictors of survival helps optimize outcomes, while addressing psychosocial challenges is crucial to minimizing the negative impact on QoL. This review highlights the need for a balanced approach to maximize the benefits of HPN.

## 1. Introduction

Home parenteral nutrition (HPN) is a critical, life-sustaining treatment for individuals with severe gastrointestinal (GI) disorders, particularly those with chronic intestinal failure (CIF) and malignant bowel obstruction (MBO) [1]. These conditions, often caused by complex diseases such as advanced cancers or severe bowel dysfunction, make normal enteral feeding insufficient or impossible, necessitating long-term intravenous nutrition. By delivering essential nutrients directly into the bloodstream, HPN supports metabolic functions and helps patients maintain body weight, muscle mass, and overall nutritional health when the digestive tract can no longer be relied upon.

The primary aim of HPN is to prolong survival and prevent malnutrition-related complications in patients who would otherwise face severe nutritional deficits. Research indicates that HPN significantly benefits survival rates for patients, especially when initiated early in the disease process and carefully monitored. Studies have shown that specific clinical indicators, including serum albumin levels, ECOG performance status, and hemoglobin levels, play a crucial role in determining patient outcomes, making patient selection and personalized HPN protocols essential for optimizing benefits [2,3].

However, while HPN can extend life, its long-term impact on patients’ quality of life (QoL) is complex and multidimensional. Factors such as reduced mobility due to dependence on infusion equipment, increased social isolation, and the psychological burden of managing a chronic condition all contribute to QoL challenges. Patients and caregivers alike often face emotional and social difficulties, including anxiety, depression, and the need for intensive support networks. For this reason, the integration of psychosocial care and tailored patient education into HPN management has been suggested as essential to maximize quality of life [4,5,6,7,8].

This review synthesizes findings from 34 studies, offering a comprehensive evaluation of HPN’s role in both survival and QoL outcomes. It identifies key clinical predictors that influence HPN outcomes and addresses the diverse physical, social, and psychological dimensions involved. By examining both the advantages and challenges of HPN, this review seeks to provide a balanced perspective on its application in advanced disease care and highlight strategies that may enhance patient and caregiver experiences with this complex, yet essential, therapy.

## 2. Methodology

A comprehensive literature search was conducted in the PubMed, Medline, and Cochrane databases, focusing on studies published from 2000 to 2024. Keywords included “home parenteral nutrition”, “chronic intestinal failure”, “quality of life”, “survival”, “oncology”, “malignant bowel obstruction”, “nutrition”. The review included prospective and retrospective cohort studies, systematic reviews, and randomized controlled trials. All the studies meeting the above criteria have been included in this review. A total of 37 studies were analyzed, with a primary focus on survival, QoL, and clinical predictors.

## 3. Impact of HPN on Survival

HPN has been shown to significantly prolong survival in patients with MBO and CIF. Dzierżanowski and Sobocki found that patients receiving HPN for MBO had a median survival of 89 days. Survival was closely linked to the patient’s performance status, with those scoring ECOG 0 or 1 having better outcomes. Serum albumin levels and the presence of water retention were also important prognostic factors [1].

Vashi et al. conducted a meta-analysis on patients with advanced cancer receiving HPN and found that the median survival was extended to 83 days, with the greatest benefit observed in patients who continued chemotherapy [2]. Similarly, Bozzetti et al. demonstrated that HPN in conjunction with chemotherapy significantly improved survival in cancer patients [3].

Naghibi et al. analyzed survival in patients with inoperable MBO and found that HPN prolonged survival by an average of 83 days. These findings were consistent across multiple studies, highlighting the role of HPN in stabilizing nutritional status and extending survival [1,2,3,9].

Several studies, including Cotogni and Fu, also emphasized the critical role of chemotherapy in improving survival. In their respective studies, patients receiving both chemotherapy and HPN experienced significantly longer survival compared to those who discontinued chemotherapy [10,11].

## 4. Impact of HPN on Quality of Life

While HPN is beneficial for survival, its effect on QoL varies greatly depending on several factors. Winkler et al. explored the QoL of patients on long-term HPN and found that patients often experience reduced physical and emotional functioning. Common issues include depression, dependence on caregivers, and disruption of social activities [4,5,6].

Sowerbutts et al. conducted a systematic review on the QoL of patients and caregivers receiving HPN. Their findings indicated that the frequency of infusions significantly impacted QoL, with fewer weekly infusions resulting in better physical and emotional outcomes [12]. This was also supported by Stanner et al. and Jones et al., who found that reducing infusion frequency improved overall well-being [7,13].

In pediatric populations, Tran et al. found that while HPN allowed children with intestinal failure to remain at home, it also introduced significant challenges, including dependency on caregivers and limitations in school attendance and social activities [14,15]. These findings were consistent across studies in both adult and pediatric populations, with the overall burden of HPN posing a significant challenge to QoL [4,5,6,7,12].

Other study led by French et al., explored the benefits of supportive interventions, such as telemedicine and psychological support, which helped improve patient satisfaction with HPN and mitigate some of the psychosocial challenges [16].

Patients on HPN face complex challenges that affect both their mental and physical well-being. Bond et al. emphasize that long-term HPN patients experience a high degree of dependence on caregivers, which can limit independence, mobility, and overall freedom. This reliance often results in social isolation and difficulties maintaining relationships and social activities, impacting emotional well-being [17].

Ablett et al. add that patients frequently report feelings of loneliness and limited daily life participation due to the rigid routines required by HPN. The structured nature of daily infusions and medical protocols restricts patients’ ability to travel, work, or engage in social functions, which can lead to frustration and decreased life satisfaction [18].

Clement et al. extend this by noting that, while HPN improves clinical outcomes in some patients, it may exacerbate social challenges, as patients feel physically constrained by the need for regular infusions. For cancer patients, these limitations are often compounded by the mental toll of their illness, potentially leading to depression or anxiety [19].

The psychological burden of HPN, therefore, emphasizes the need for integrated psychosocial support, which can alleviate some emotional strain and improve coping skills. As studies show, providing access to social workers, psychologists, and supportive communities could positively influence the quality of life for HPN patients by fostering resilience and offering practical assistance.

The use of HPN for terminally ill patients raises ethical questions, particularly when treatment extends life without necessarily enhancing quality of life. Emanuel et al. discuss the importance of respecting patient autonomy and wishes in HPN decision making. Since HPN can be both a lifeline and a burden, it is crucial to assess whether treatment aligns with the patient’s values and goals. In cases where HPN may prolong suffering, health professionals are encouraged to adopt a patient-centered approach, weighing the benefits against potential distress and preserving dignity. Additionally, ethical considerations include transparent communication with patients and families regarding the potential outcomes and limitations of HPN. Clinicians must carefully navigate the fine line between offering life-sustaining care and avoiding unnecessary interventions that might lead to a diminished quality of life [20].

Ma et al. contributes a vital perspective by showing that HPN can sustain the quality of life in patients with incurable gastric cancer undergoing salvage chemotherapy. In this study, HPN helped maintain body weight, stabilized nutritional parameters (e.g., protein, prealbumin, and cholesterol), and preserved global quality of life (QoL) scores comparable to those in the control group. These findings suggest that HPN, when introduced early, can support both nutritional health and QoL, even during aggressive treatments [21].

Bohnert et al. showed that while HPN contributes positively by stabilizing physical health and nutritional status, it often restricts daily activities, leading to limited mobility and dependence on structured routines. These limitations can contribute to feelings of social isolation and frustration, as patients are often unable to fully participate in previously enjoyed activities. This study highlights the dual effect of HPN, where physical benefits are tempered by significant social impacts [22].

Reber et al. further underscore the need for comprehensive psychosocial support in HPN care, showing that long-term HPN often leads to emotional distress and social isolation. According to their study, patients frequently report feelings of loneliness and emotional burden, indicating that HPN programs must incorporate access to social workers, counseling, and community resources to help patients cope. This psychosocial support could enhance resilience and provide a more balanced experience for individuals relying on HPN [23].

In conclusion, these studies [22,23] collectively underscore the importance of addressing not only the physical health but also the emotional and social needs of HPN patients. While HPN provides critical life-sustaining benefits, it is clear that a more integrated support system—including mental health resources and social support—can play a key role in improving overall quality of life for patients.

Kirk et al. examined the perspectives of healthcare providers in the UK regarding health-related quality of life (QoL) assessments for patients on HPN, highlighting both the importance and challenges of implementing QoL evaluations. The study revealed that although healthcare professionals recognize the value of assessing QoL to tailor care, routine implementation of these assessments is limited, with inconsistencies in both frequency and methodology across providers. This lack of standardized monitoring may lead to missed opportunities for early intervention in addressing patient needs. It was suggested that integrating regular, structured QoL assessments into HPN care would allow for a more personalized approach, helping providers identify and respond to social, emotional, and physical challenges faced by patients [24].

Hu et al. evaluated the quality of life (QoL) of adult patients on home parenteral nutrition (HPN) in North East England and Cumbria. The authors used the HPN-QOL (Home Parenteral Nutrition—Quality of Life) assessment tool, which measures various aspects of functioning and symptoms that may affect patients’ quality of life on long-term HPN. Patients rated their ability to travel, physical functioning, employment, and sexual function as poor. Fatigue emerged as a primary limiting factor, having a significant impact on overall quality of life. The study also explored differences across age and gender. Male patients reported better scores for nutritional intake and support from healthcare teams but experienced more gastrointestinal symptoms. Patients over the age of 55 reported lower employment scores and more frequent gastrointestinal issues. The study suggests that QoL should be considered an integral part of clinical care for HPN patients, as many face challenges that affect their daily functioning and life satisfaction. Addressing these QoL factors may provide a more comprehensive and supportive approach to managing long-term HPN [8].

The study by Saqui et al. evaluated the impact of transitioning from stationary to portable infusion pumps on the QoL of HPN patients in Canada. It concludes that portable pumps offer significant QoL advantages over stationary models, particularly in reducing sleep disruption, increasing social flexibility, and enhancing overall satisfaction. The authors suggest that these QoL improvements justify the use of portable pumps for HPN patients, as they offer a meaningful enhancement to daily life and independence [25].

## 5. Clinical Predictors of HPN Outcomes

Several clinical and laboratory markers have been identified as critical predictors of HPN outcomes:

Performance status (ECOG): Numerous studies show that patients with an ECOG score of 0 or 1 are more likely to benefit from HPN, while those with higher scores have significantly shorter survival [1,2,3].

Serum albumin: Hypoalbuminemia (<2.5 g/dL) is consistently associated with poor outcomes, indicating severe malnutrition [1,2].

Hemoglobin: Anemia, even in mild forms, is linked to reduced survival, making it a key factor in determining patient eligibility for HPN [1,2].

Water retention: Patients with peripheral edema, ascites, or hydrothorax are less likely to benefit from HPN, with studies showing a significantly shorter survival in these populations [1].

Santarpia et al. examined predictive factors of survival in patients with MBO on HPN. Their findings indicated that low serum albumin levels and severe malnutrition were strongly associated with reduced survival. Patients with higher albumin levels and better overall nutritional status (e.g., absence of edema and other complications) demonstrated longer survival rates on HPN. This study emphasizes the importance of regular nutritional assessments, as maintaining higher albumin levels appears to be a critical factor for achieving better outcomes in patients on long-term HPN [26].

Ma et al. focused on identifying factors that predict survival in patients with incurable gastric cancer receiving HPN. This study highlighted that patients with a better Eastern Cooperative Oncology Group (ECOG) performance status (0–1), higher albumin levels, and improved nitrogen balance had significantly longer survival compared to those with poor ECOG scores and low albumin levels. The study underscores the importance of considering ECOG status and serum albumin as essential indicators in patient selection for HPN, as these parameters are closely associated with both survival and quality of life [21].

These findings contribute to a more comprehensive understanding of how specific clinical indicators—particularly serum albumin levels, ECOG performance status, and overall nutritional balance—impact the outcomes of patients on HPN. By closely monitoring these factors, healthcare providers can better assess which patients are likely to benefit from HPN and adjust treatment protocols to optimize results. Integrating such predictive markers into HPN management plans can significantly enhance patient outcomes, especially for those with advanced cancers and complex nutritional needs.

## 6. Clinical Outcomes of HPN

HPN has demonstrated substantial clinical benefits in stabilizing health, improving survival outcomes, and maintaining nutritional status in patients with severe intestinal failure and advanced cancer.

Pinto-Sanchez et al. focused on patients with short bowel syndrome (SBS) receiving HPN, a population at high risk of malnutrition due to limited absorptive capacity. Their study found that HPN significantly improved clinical outcomes, including stabilization of body weight, BMI, and vital nutritional parameters. HPN also helped prevent hospitalizations related to malnutrition, showcasing its role in maintaining health and reducing the need for acute medical interventions [27].

Senesse et al. explored HPN’s benefits in patients with advanced cancer and intestinal failure. They observed that HPN not only improved nutritional status but also contributed to increased survival and reduced malnutrition-related complications. Cancer patients receiving HPN had greater tolerance to chemotherapy, suggesting that nutritional support through HPN may enhance treatment efficacy and improve overall clinical outcomes in oncology [28].

Cotogni et al. conducted a retrospective study on cancer patients receiving HPN, noting that it significantly stabilized their nutritional status and provided key support during cancer treatment. This study found that patients on HPN had fewer hospital admissions for complications related to malnutrition and were better able to withstand the side effects of intensive cancer therapies. These findings highlight HPN as an effective tool in enhancing the resilience of cancer patients and helping them manage the challenges of long-term treatment [10].

Bohnert et al. examined long-term clinical outcomes for patients on HPN, noting significant improvements in both survival and overall health status. The study underscored HPN’s role in preventing deterioration in severely malnourished patients, particularly those with chronic or incurable conditions. By enabling patients to meet their nutritional needs at home, HPN reduced the frequency of hospitalizations, thus improving quality of life and allowing patients to remain in a more comfortable, familiar setting [22].

Reber et al. conducted a multicenter study that analyzed long-term HPN outcomes across various patient groups. Their findings indicated that HPN contributed to a sustained improvement in nutritional parameters, leading to better functional status and, in many cases, extended survival. The study emphasized the importance of personalized HPN protocols to address individual patient needs, optimizing both clinical outcomes and patient satisfaction [23].

Collectively, these studies underscore the essential role of HPN in supporting patients with complex nutritional requirements. For those with conditions such as SBS, advanced cancer, or other forms of intestinal failure, HPN provides a stable, sustainable source of nutrition, enhancing resilience, supporting treatment tolerability, and contributing to prolonged survival. Through effective HPN management, patients not only experience improved physical health but also gain greater independence and reduced reliance on hospital-based care, significantly enhancing their overall clinical outcomes.

A study by Van Gossum analyzed clinical outcomes for patients receiving HPN following bariatric surgery complications, such as anastomotic leaks, fistulas, severe protein deficiency (hypoalbuminemia), and vitamin deficiencies. The findings highlight that while HPN was effective in stabilizing nutritional status, it was primarily a temporary solution, often used as a “bridge” therapy until further surgical interventions could be performed.

Key Results:A high rate of rehospitalization (58%) was observed among patients on HPN, underscoring the complex needs and challenges faced by this population.Vascular complications, such as catheter-related issues, affected 41% of patients, indicating a significant risk associated with long-term HPN in these cases.

The study concludes that HPN provides critical support for patients with severe nutritional deficits due to postoperative complications; however, it is not without substantial risks. These findings suggest that HPN should be carefully managed, with attention to preventing complications and preparing for potential surgical solutions as patients stabilize. This research underscores the importance of assessing each patient’s unique clinical factors, such as protein deficiency and post-surgical complications, to optimize HPN outcomes and anticipate possible rehospitalizations [29].

Lezo et al. [30], in their paper, provide valuable insights into long-term outcomes and the standard of care for pediatric patients on HPN due to CIF over a 28-year period in an Italian reference center. It was found that the survival rate was high among pediatric patients, with a notable percentage achieving enteral autonomy over time. For example, 74.5% of patients were weaned off HPN within two years, while dependence rates dropped significantly over time. The study also showed a decrease in catheter-related bloodstream infections (CRBSI) over the years, from 0.33 to 0.19 episodes per 1000 catheter days after implementing a taurolidine lock solution in 2011. Although QoL was not formally measured, data indicated that 90% of patients attended school, and 81% could go on holidays outside their region, with 48% participating in sports. Additionally, 67% of caregivers, mostly mothers, remained employed, suggesting a well-managed daily life despite HPN dependency. The study underscores the importance of standardized care protocols for pediatric HPN patients with CIF, which contribute to high survival rates, reduced complications, and improved quality of life. The data indicate that specialized HPN centers with multidisciplinary teams are essential in managing CIF effectively and enhancing both clinical and quality of life outcomes for pediatric patients.

Similar observations were made by Lowthian et al. and Meijerink et al. They postulate for a re-evaluation of HPN management to prioritize quality of life outcomes, suggesting that HPN care should go beyond merely sustaining life and instead focus on enhancing overall well-being. This includes better patient education, enhanced social support, and developing standardized outcome measures for quality of life to ensure that care aligns with the needs and values of patients [31,32].

## 7. Role of Chemotherapy and Other Interventions

The continuation of chemotherapy is a key factor in improving outcomes for patients on HPN. Dzierżanowski and Sobocki found that patients who continued chemotherapy while receiving HPN had significantly higher survival rates compared to those who discontinued chemotherapy [1]. Similarly, Cotogni and Dashti et al. emphasized the importance of integrating HPN with chemotherapy to maximize survival [10,33,34].

Additionally, psychological support, as highlighted by Senesse et al., were found to enhance QoL by providing additional resources for managing the challenges of HPN [28].

## 8. Limitations of Current Research and Future Directions

This review highlights the need for continued research on home parenteral nutrition (HPN) to enhance patient selection, quality of life (QoL), and treatment efficacy. Future studies should focus on:Standardized QoL Assessment—develop and validate HPN-specific QoL tools (e.g., HPN-QoL, SBS-QoL) to improve cross-study comparisons and patient care.Personalized Patient Selection—use predictive models to refine selection criteria, ensuring HPN benefits those most likely to improve.Psychosocial Support—integrate telemedicine, support groups, and mental health interventions to mitigate the emotional burden of HPN.Optimized Infusion Schedules—explore cyclic HPN and portable infusion technologies to enhance mobility and daily functioning.Long-Term Outcomes and Cost-Effectiveness—assess the economic impact, hospitalizations, and alternatives like gut rehabilitation or intestinal transplantation.Oncology and Palliative Care Integration—determine HPN’s role in chemotherapy support, ethical considerations, and criteria for discontinuation.Technological Advancements—develop AI-driven monitoring, smart infusion systems, and personalized nutrition formulations for better patient outcomes.

Future research should move beyond survival benefits, focusing on holistic patient care, improved QoL, and innovative treatment approaches to make HPN a more effective and sustainable therapy.

## 9. Conclusions

HPN is an indispensable intervention for patients with CIF and advanced cancers, particularly those who face significant challenges in achieving adequate nutrition through enteral feeding. This therapy provides critical nutritional support, helping to prevent malnutrition and extend survival for patients in both palliative and supportive care settings. Research consistently shows that HPN can stabilize or improve nutritional markers, such as albumin and hemoglobin levels, which are strongly associated with better clinical outcomes. For certain patients, especially those with favorable clinical indicators (e.g., ECOG performance scores of 0–1), HPN can sustain health and quality of life while enhancing the effectiveness of concurrent treatments like chemotherapy.

However, the impact of HPN on QoL remains multifaceted and often varies widely across individuals. QoL is influenced not only by the physical aspects of HPN, such as dependence on infusion devices and the risk of complications (e.g., catheter-related infections), but also by significant psychosocial and lifestyle factors. Studies, including those by Bohnert et al. [22] and Reber et al. [23] highlight challenges such as social isolation, decreased mobility, and emotional burdens. These findings underscore the need for a holistic approach to HPN, one that combines medical management with comprehensive psychosocial support systems, including mental health services, social networks, and educational resources for both patients and caregivers.

Key clinical predictors, including performance status, serum albumin levels, and the presence of water retention, have emerged as crucial factors for assessing the suitability and potential benefit of HPN. Patients who continue chemotherapy alongside HPN tend to have better survival outcomes, suggesting that HPN may help sustain their tolerance to aggressive cancer treatments.

Moving forward, a more personalized approach to HPN is essential, with ongoing research aimed at refining patient selection criteria to ensure that those who are most likely to benefit from HPN receive this support. Future studies should also explore the long-term psychosocial effects of HPN on both patients and caregivers, focusing on strategies to mitigate QoL challenges and improve patient satisfaction. Furthermore, advances in portable infusion technology and integrated care models, as indicated by Jones et al. and Hu et al. [8,13], have the potential to enhance patient autonomy and reduce the burden on daily life, thereby fostering a better overall experience with HPN.

Moving forward, HPN research must shift from a survival-focused approach to a patient-centered model that prioritizes personalized treatment, psychosocial well-being, and technological innovations. By integrating advanced predictive analytics, QoL-enhancing interventions, and ethical considerations, HPN can be optimized to not only sustain life but also improve the overall well-being of patients.

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
