# Peer review of "The Impact of Home Parenteral Nutrition on Survival and Quality of Life in Patients with Intestinal Failure and Advanced Cancer: A Comprehensive Review"

_nutrients, 2025, doi:10.3390/nu17050905_

Round 1
Reviewer 1 Report
Comments and Suggestions for Authors
Journal: Nutrients
Section
Clinical Nutrition
Special Issue
Nutrition and Quality of Life for Patients with Chronic Disease
Article title:
The impact of home parenteral nutrition on survival and quality of life in patients with intestinal failure and advanced cancer: A comprehensive review.
Dear Editor,
Thank you for inviting me to review this manuscript submitted to Nutrients.
OVERALL COMMENTS
Based on the statement that “Home parenteral nutrition (HPN) is essential in the management of chronic intestinal failure (CIF) and malignant bowel obstruction (MBO)…“, the authors of this study intended to the evidence from studies that evaluated the impact of HPN on survival and quality of life (QoL) in patients with MBO, CIF, and advanced cancer, as well as to identify clinical predictors of survival and address psychosocial challenges. Their results showed that the key predictors of improved survival included good performance status, higher albumin levels, and the ability to continue chemotherapy. They found tha t psychosocial challenges, including lifestyle disruption and mental health strain, were common among long-term HPN patients. In conclusion, the authros point that HPN can be a life-sustaining therapy for patients with CIF, MBO, and advanced cancer, but its success depends on careful patient selection and management.
TITLE
The title is adequate.
_______
ABSTRACT
This section is also adequate.
_______
KEYWORDS
The included keywords were were: home parenteral nutrition; quality of life; survival; oncology; malignant bowel obstruction; nutrition.
I suggest: home parenteral nutrition; chronic intestinal failure; quality of life; survival; oncology; malignant bowel obstruction; nutrition
_______
INTRODUCTION
I can understand that the authors of this article may have a deep knowledge of the subject and therefore wrote a good introduction. However, it would be more acceptable if numerous references were included that would provide support for the added information, even if they were articles published by the authors of this review. Please include references published mainly in the last 3 years.
_______
METHODS
The keyword used to perform the search in the chosen databases are adequate ("home parenteral nutrition," "malignant bowel obstruction," "chronic intestinal failure,""quality of life," and "survival"), but I also suggest including: “oncology” and “nutrition”
____________
In the sections:
3. Impact of HPN on survival
4. Impact of HPN on quality of life
5. Clinical predictors of HPN outcomes
6. Clinical outcomes of HPN
6. Clinical outcomes of HPN
I suggest that authors check the size of the font utilized and the references citation. Please double-check MDPI guidelines. References should be cited in the text in numbers in brackets and not (). They also are not cited in bold. Please check.
The definition of some abbreviations is necessary. For example, please define HPN-QoL and SBS-QoL.
LIMITATIONS OF CURRENT RESEARCH AND FUTURE DIRECTIONS
Please expand the Future Perspectives of this study. How can this review contribute to further research?
_________
CONCLUSION
This section is adequate. However, I suggest expanding it in future perspectives.
Please check MDPI guidelines and see the references cited in this section.
REFERENCES
Please review MDPI guidelines.
Please see lines 408-409. There is something wrong because reference 17 turned 1.
Author Response
Dear Reviewer, Thank you very much for the provided comments. All of them have been taken into account.
Reviewer 2 Report
Comments and Suggestions for Authors
The manuscript offers a comprehensive review of the role of home parenteral nutrition (HPN) in the treatment of chronic intestinal failure (CIF), malignant bowel obstruction (MBO), and advanced cancer. It assesses survival outcomes, the impact on quality of life (QoL), and clinical predictors, drawing on evidence from 38 studies.
1. Provide more details on how studies were selected, including inclusion and exclusion criteria.
2.Add table summarising the data from the studies.
3.Add a table with the key findings of the studies.
4. Give an algorithm for managing these patients.
Author Response
Thank yo for your really useful comments. Most of them we have implemented to this paper.
Thank you.
Reviewer 3 Report
Comments and Suggestions for Authors
I would like to begin by congratulating the authors on the theme of this review, “The impact of home parenteral nutrition on survival and quality of life in patients with intestinal failure and advanced cancer: a comprehensive review.” I consider the topic relevant and of great interest to health professionals and the scientific community, taking into account the difficulties of research in patients with home parenteral nutrition, since it is very difficult to control all the variables. I would like to highlight that the review emphasized the importance of personalized HPN protocols to meet the individual needs of patients, optimizing clinical results and patient satisfaction, and the essential role of HPN in supporting patients with complex nutritional needs.
I would like to make some comments and questions that I believe may be a contribution to this review:
Introduction
Paragraph 43……….. specific clinical indicators, including serum albumin levels, ……….. And not pre-albumin?
Methodology
- I missed a more detailed methodology; what were the exclusion criteria?
- In my opinion, they could have used a specific methodology, such as PRISMA.
- Don't you think it's important to know the content of the parenteral nutrition formulas used in the various studies?
- The clinical predictors should be described here, as well as the cutoff points, as well as the questionnaires used to determine quality of life, because there were several.
Paragraph 82………in the stabilization of nutritional status…. Wouldn't it have been important to highlight the type, caloric value, and output of parenteral nutrition? Were complications not assessed in the studies? Namely refeeding syndrome?
Paragraph 201………Serum albumin: Hypoalbuminemia is consistently associated with poor results, indicating severe malnutrition…. And pre-albumin? Wouldn't it be a much more sensitive and specific marker?
Paragraph 230………advanced and complex nutritional needs……….And very specific ones that should always be individualized!
Paragraph 323………..Limitations of current research and future directions……….. I agree with these limitations, but would also add that only studies with elderly and adults should have been selected and not with children.
Conclusions
Paragraph 336 ……Research consistently shows that HPN can stabilize or improve nutritional markers, such as albumin and hemoglobin levels…. What about prealbumin?
Author Response
Dear Reviewer, Thank you so much for your engagement and input on this paper. Most of your comments were taken under consideration. In case of prealbumin - We have analysed accesible data and prealbumin were not mentioned there. I agree that this parameter is more sensitive (half time is much more shorter than in case of ALB), but on the other hand there are more confounding factors to influence the final results.
Highlighting the type, caloric value, and output of parenteral nutrition - Our intention was to present parenteral nutrition as a treatment method, regardless of the type of product administered.
Thank you
Round 2
Reviewer 1 Report
Comments and Suggestions for Authors
Dear authors,
Thank you for making the corrections to the manuscript.
I wish you good luck.
Author Response
Comment:Thank you for making the corrections to the manuscript.
I wish you good luck.
Response:Thank you for taking the time to give us your professional comments.
Reviewer 2 Report
Comments and Suggestions for Authors
The authors did not make the necessary revisions according to the provided suggestions.
Author Response
Comment 1:Provide more details on how studies were selected, including inclusion and exclusion criteria.
Response: “A comprehensive literature search was conducted in PubMed, Medline, and Cochrane databases, focusing on studies published from 2000 to 2024. Keywords included “home parenteral nutrition”, “chronic intestinal failure”, “quality of life”, “survival”, “oncology”, “malignant bowel obstruction”, “nutrition". The review included prospective and retrospective cohort studies, systematic reviews, and randomized controlled trials. All the studies meeting the above criteria have been included in this review. A total of 38 studies were analyzed, with a primary focus on survival, QoL, and clinical predictors.”
Please refer to lines 63-70 in the manuscript.
Comment 2:Add table summarising the data from the studies.
Response: Please refer to Table S1 in the Supplementary Material
Comment 3:Add a table with the key findings of the studies.
Response: Please refer to Table S1 in the Supplementary Material
Comment 4: Give an algorithm for managing these patients.
Response: An algorithm for managing patients: The absence of a standardized algorithm for managing patients with intestinal failure and advanced cancer is a result of multiple barriers, including patient heterogeneity, inconsistent evidence, diverse clinical approaches, ethical dilemmas, and logistical challenges. Future research should focus on developing personalized treatment strategies, incorporating both clinical and psychosocial factors, to optimize patient care. We hope that this response clarifies the rationale behind our findings and appreciate the reviewer's thoughtful feedback on this critical issue.